

# Screening of faba bean (*Vicia faba* L.) accessions to acidity and aluminium stresses

Kiflemariam Y. Belachew[1] and Frederick L. Stoddard[2]

[1] Department of Agricultural Sciences, Viikki Plant Science Centre, University of Helsinki, Helsinki, South Finland, Finland

[2] Department of Food and Environmental Sciences, Viikki Plant Science Centre, University of Helsinki, Helsinki, Finland

## ABSTRACT

**Background**. Faba bean is an important starch-based protein crop produced worldwide. Soil acidity and aluminium toxicity are major abiotic stresses affecting its production, so in regions where soil acidity is a problem, there is a gap between the potential and actual productivity of the crop. Hence, we set out to evaluate acidity and aluminium tolerance in a range of faba bean germplasm using solution culture and pot experiments.

**Methods**. A set of 30 accessions was collected from regions where acidity and aluminium are or are not problems. The accessions were grown in solution culture and a subset of 10 was grown first in peat and later in perlite potting media. In solution culture, morphological parameters including taproot length, root regrowth and root tolerance index were measured, and in the pot experiments the key measurements were taproot length, plant biomass, chlorophyll concentration and stomatal conductance.

**Result**. Responses to acidity and aluminium were apparently independent. Accessions Dosha and NC 58 were tolerant to both stress. Kassa and GLA 1103 were tolerant to acidity showing less than 3% reduction in taproot length. Aurora and Messay were tolerant to aluminium. Babylon was sensitive to both, with up to 40% reduction in taproot length from acidity and no detectable recovery from $Al^{3+}$ challenge.

**Discussion**. The apparent independence of the responses to acidity and aluminium is in agreement with the previous research findings, suggesting that crop accessions separately adapt to $H^+$ and $Al^{3+}$ toxicity as a result of the difference in the nature of soil parent materials where the accession originated. Differences in rankings between experiments were minor and attributable to heterogeneity of seed materials and the specific responses of accessions to the rooting media. Use of perlite as a potting medium offers an ideal combination of throughput, inertness of support medium, access to leaves for detection of their stress responses, and harvest of clean roots for evaluation of their growth.

Corresponding author
Kiflemariam Y. Belachew,
kiflemariam.belachew@helsinki.fi

## INTRODUCTION

Growth and yield of faba bean are determined by climatic, edaphic, and management practices that are not independent of each other and interact to affect the chemical characteristics of the soil. Soil acidity has a dramatic impact on most chemical and biological processes of a crop. Faba bean (*Vicia faba* L.) grows best in soils with pH ranging from 6.5 to 9.0 (*Jensen, Peoples & Hauggaard-Nielsen, 2010*), and is considered to fare poorly at a pH values of 5 or less (*French & White, 2005*). Nevertheless, some accessions remain productive when soil pH is as low as 4.5 (*Singh et al., 2012*), with a critical soil pH value of 4.0 below which assimilation of major ions decreases (*Schubert, Mengel & Schubert, 1990*), and net $H^+$ release and root growth cease (*Yan, Schubert & Mengel, 1992*).

Soil acidification is a worldwide problem, and sensitivity to acid soils limits the usage of faba bean in some cropping systems. Besides the simple matter of low pH, soil acidity is associated with high availability of $Al^{3+}$, which is stressful or toxic to many plants. Acid soil can be managed by the application of lime and the effect of $Al^{3+}$ toxicity is ameliorated by the use of P-containing fertilizers (*Liao et al., 2006*; *Atemkeng et al., 2011*). However these options are not available where farmers are poor or legislation restricts fertilizer usage, they are less effective when cultivars are sensitive (*Sun et al., 2008*), and acidity in subsoil is harder to address than that in topsoil (*Hede, Skovmand & Lopez-Cesati, 2001*; *Brown et al., 2008*; *Zheng, 2010*). Furthermore, in areas as diverse as sub-Saharan Africa and Western Australia, soil acidity has resulted in changes of cropping sequence. In Ethiopia, faba bean has been abandoned in some agricultural regions where it was previously part of the cropping system and has not been replaced by another food legume (*Genanew, Argaw & Adgo, 2012*). In areas where soil acidity is severe, cereal rye cropping and livestock grazing may be the only remaining options for farmers (*Genanew, Argaw & Adgo, 2012*).

Most of the studies on the inheritance of $Al^{3+}$ resistance have been conducted on the major cereal crops (wheat (*Triticum aestivum* L. emend Thell), rice (*Oryza sativa* L.), and maize (*Zea mays* L.)) (*Kochian, Piñeros & Hoekenga, 2005*), a few legumes such as soybean (*Glycine max* (L.) Merr.) (*Liao et al., 2006*), pigeon pea (*Cajanus cajan* L.) (*Choudhary, Singh & Kumar, 2011*), and barrel medic (*Medicago truncatula* L.) (*Narasimhamoorthy et al., 2007*; *Chandran et al., 2008*). Studies on this trait in faba bean have been few in number. Nevertheless, wide diversity exists among faba bean landraces for agro-ecological adaptation (*French & White, 2005*; *Wondafrash, 2006*) and biotic and abiotic stress resistance (*Stoddard et al., 2006*; *Shifa, Hussien & Sakhuja, 2011*; *Khazaei et al., 2013*). $Al^{3+}$ toxicity tolerance of cv 'Herz Freya' was lower than that of barley and much lower than in yellow lupin and rye (*Horst & Göppel, 1986a*; *Horst & Göppel, 1986b*).

To identify sources of resistance to acidity and aluminium stresses, an efficient screening method is required. Solution culture, where seedlings are suspended with their roots in an aerated nutrient solution of known $Al^{3+}$ concentration, is suitable for screening of tens or hundreds of accessions, and has proven useful in wheat (*Stodart et al., 2007*), pigeon pea (*Choudhary, Singh & Kumar, 2011*; *Choudhary & Singh, 2011*), barley (*Hordeum vulgare* L.) (*Nawrot, Szarejko & Maluszynski, 2001*; *Echart et al., 2002*; *Tamas et al., 2006*), sorghum (*Sorghum bicolor* L.) (*Hill, Ahlrichs & Ejeta, 1989*), and barrel medic (*Narasimhamoorthy*

*et al., 2007*). Selected germplasm can then be tested in soil or potting mix, so differences in growth and yield can be determined. Staining of root tips with hematoxylin, measurement of root regrowth after transfer to non-stress conditions, and relative root growth as a measure of root tolerance index after exposing plant roots to toxic levels of $Al^{3+}$ are some of the techniques that are employed (*Hede, Skovmand & Lopez-Cesati, 2001*) and allow simple ranking of tolerance and sensitivity (*Choudhary & Singh, 2011*). Reliable results are, however, most likely to be obtained by the application of multiple procedures (*Narasimhamoorthy et al., 2007*).

Other morpho-physiological measures may also be informative about aluminium and acidity responses in faba bean. Leaf gas exchange was reduced by 2–3 fold in tomato cultivars as aluminium concentration increased to 50 µmol/l (*Simon et al., 1994*). Similarly, 0.1 mM aluminium reduced root weight in winter wheat (*Szabó-Nagy, 2015*) and at pH 4, aluminium concentration of 0.13–0.15 mmol/kg significantly reduced photosynthesis, chlorophyll concentration and transpiration in wheat (*Ohki, 1986*). Other ionic stresses such as salinity increased chlorophyll a fluorescence level in both mung bean and Brassica seedlings (*Misra, Srivastava & Strasser, 2001*).

For these reasons, we set out to evaluate acidity and aluminium tolerance in a range of faba bean germplasm. Solution culture of seedlings and pot experiments on pre-reproductive plants were both used in order to develop a reliable technique for discriminating sensitive and tolerant germplasm.

## MATERIALS AND METHODS

### Plant materials

Thirty faba bean accessions were chosen for this study on the basis of their expected exposure to acidity or aluminium stress in their regions of provenance (Table 1). One accession had to be dropped due to inconsistent germination during the experiment. Twenty Ethiopian accessions were provided from the germplasm collection of the Ethiopian Institute of Agricultural Research, Holeta Agricultural Research Center (HARC) and the remainder were chosen from European and Canadian germplasm used in previous experiments. Acid soils are found in the wet highlands of Ethiopia where faba bean is predominantly grown. Acid soils occupy 41% of the country (*Abebe, 2007*), and of this area nearly one-third has an aluminium toxicity problem (*Schlede, 1989*). The first solution culture experiments were conducted with seed materials as received, the first pot experiment with material that had been inbred for one generation, and the second experiment with seeds from a single third-generation inbred plant of each accession.

### Screening for tolerance to acidity and aluminum in solution culture
#### *Plant growing conditions*

Seeds of uniform size were selected from each of the accessions, washed three times in tap water, disinfected with 1% NaClO (sodium hypochlorite) (w/v) for 5 min and rinsed 3 times with running tap water. The seeds were soaked in tap water for 24 h, transferred to three layers of moist filter paper in 14 cm diameter Petri dishes (14–20 seeds/dish), and incubated for 72 h at 22 °C in the dark. The seedlings were then transferred to holes
**Table 1  The 29 accessions of faba bean, their country of origin and source, and their response to 82 μmol/l Al³⁺ in solution culture, n = 3.**

| Accession | | | Aluminium responses | | | | | | |
|---|---|---|---|---|---|---|---|---|---|
| | Origin | | Haematoxylin stain | | Root regrowth | | Al root tolerance index (82 μmol/l Al³⁺/ 0 μmol/l Al³⁺) | | |
| | Country | Source | Score | Rank | Length (cm) | Rank | Value | Rank | Rank sum |
| Alexia | Austria | Gleisdorf | 1.5 | 14 | 0.2 | 2 | 1.04 | 21 | 37 |
| Aurora | Sweden | Svalöf Weibull | 1.3 | 20 | 0.8 | 20 | 1.14 | 25 | 65 |
| Babylon | Netherlands | Nickerson Limagrain | 2.8 | 1 | 0.0 | 1 | 0.98 | 15 | 17 |
| Bulga 70 | Ethiopia | HARC | 1.8 | 9 | 0.3 | 5 | 0.97 | 13 | 27 |
| CS 20 DK | Ethiopia | HARC | 1.9 | 7 | 0.2 | 4 | 1.03 | 19 | 30 |
| Degaga | Ethiopia | HARC | 2.4 | 2 | 0.4 | 8 | 1.14 | 26 | 36 |
| Divine | France | INRA | 1.7 | 12 | 0.4 | 9 | 0.82 | 4 | 25 |
| Dosha | Ethiopia | HARC | 0.8 | 24 | 1.3 | 26 | 0.95 | 12 | 62 |
| EH 06006-6 | Ethiopia | HARC | 1.7 | 10 | 0.3 | 6 | 1.47 | 29 | 45 |
| EK 02016-1 | Ethiopia | HARC | 1.6 | 13 | 0.8 | 18 | 1.04 | 20 | 51 |
| Fatima | Canada | Univ. Saskatchewan | 2.1 | 4 | 1.3 | 25 | 0.89 | 9 | 38 |
| Gebelcho | Ethiopia | HARC | 1.0 | 23 | 0.9 | 21 | 1.07 | 22 | 66 |
| GLA 1103 | Austria | Gleisdorf | 1.8 | 8 | 0.5 | 11 | 0.99 | 16 | 35 |
| Gora | Ethiopia | HARC | 1.3 | 19 | 0.4 | 10 | 0.59 | 1 | 30 |
| Hachalu | Ethiopia | HARC | 1.4 | 17 | 1.5 | 28 | 1.03 | 18 | 63 |
| Holetta-2 | Ethiopia | HARC | 1.3 | 21 | 0.6 | 13 | 0.93 | 10 | 44 |
| Kassa | Ethiopia | HARC | 2.0 | 5 | 0.2 | 3 | 0.83 | 5 | 13 |
| Kontu | Finland | Boreal | 2.3 | 3 | 0.3 | 7 | 1.23 | 28 | 38 |
| KUSE | Ethiopia | HARC | 1.5 | 15 | 0.6 | 14 | 0.88 | 7 | 36 |
| Melodie | France | INRA | 1.4 | 16 | 1.0 | 23 | 0.79 | 3 | 42 |
| Messay | Ethiopia | HARC | 1.4 | 18 | 0.8 | 19 | 1.19 | 27 | 64 |
| Moti | Ethiopia | HARC | 1.8 | 9 | 1.0 | 24 | 1.02 | 17 | 50 |
| NC 58 | Ethiopia | HARC | 1.1 | 22 | 0.9 | 22 | 0.94 | 11 | 55 |
| OBSE | Ethiopia | HARC | 2.0 | 5 | 0.6 | 15 | 0.87 | 6 | 26 |
| SSNS-1 | Canada | Univ. Saskatchewan | 2.0 | 5 | 0.4 | 8 | 1.08 | 23 | 36 |
| Tesfa | Ethiopia | HARC | 1.7 | 11 | 0.7 | 17 | 0.89 | 8 | 36 |
| Tumsa | Ethiopia | HARC | 2.0 | 5 | 1.3 | 27 | 1.11 | 24 | 56 |
| Walki | Ethiopia | HARC | 1.5 | 14 | 0.6 | 16 | 0.77 | 2 | 32 |
| Wayu | Ethiopia | HARC | 1.9 | 6 | 0.5 | 12 | 0.97 | 14 | 32 |
| SE | | | 0.29 | | 0.23 | | | | |
| LSD (5%) | | | 0.82 | | 0.66 | | | | |

in a lid suspended over a plastic tray (78 cm × 56 cm × 18 cm) containing 0.5 mM CaSO₄ solution, and grown in the dark for another four days in controlled-environment growth chambers at 22/20 °C day/night temperatures with 70% relative air humidity. The solution was continuously aerated with an aquarium pump. Eight seedlings of each of the 29 accessions were used in each tray. Three replicates of five treatments were prepared, and arranged in a split-plot design with replicate as the block, tray (treatment) as the main plot

and accession as the subplot. The replicates were separated by 21 days in time rather than physically in space, owing to space restrictions.

For the remaining seven days of the experiment, seedlings were allowed to grow on continuously aerated nutrient solution comprising 4.0 mM $CaCl_2$, 6.5 mM $KNO_3$, 2.5 mM $MgCl_2$, 0.1 mM $(NH_4)_2SO_4$ , and 0.4 mM $NH_4NO_3$, with fresh nutrient solution culture provided on day 4. The pH was corrected to 7.0 (neutral control) and 4.5 (acid and aluminium treatments) with 10.88 M HCl and 0.5 M $Na_2CO_3$ solutions. The growing conditions were 16/8 h light/dark regime, 22/20 °C day/night temperatures, 70% relative air humidity and a photon flux density of 230 $\mu$mol m$^{-2}$ s$^{-1}$ at the plant canopy. After two days of this 7-day interval, the aluminium treatments were initiated by transferring three sets of seedlings to fresh nutrient solution containing 41, 82, and 123 $\mu$mol/l aluminum sulfate $[Al_2(SO_4)_3.16H_2O]$. One day later, four seedlings from every set of eight were removed for haematoxylin staining (see below) and after a further day, the remaining seedlings were transferred to aluminium-free solutions at pH 4.5 and allowed to recover for three days, as recommended by *Nava et al. (2006)*, bringing their total growing time to seven days after their transfer to the nutrient solution culture, the same as for those not exposed to $Al^{3+}$.

At the end of the growth period, the length of the primary root of each seedling was measured with a ruler, showing its tolerance to the stress. On the aluminium-treated seedlings, the length of the primary root from the tip to the point of root thickening (callus), caused by the aluminium treatment, was measured with a ruler and termed as root regrowth length. This measure demonstrates the ability of the plant to recover from the stress.

The protocol for staining and scoring aluminium response followed *Polle, Konzak & Kittrick (1978)* and was based on the affinity of haematoxylin for chelated aluminium. The sets of 4 seedlings were transferred to distilled water with occasional shaking for 60 min to remove aluminium from the root surface. The root tips were immersed in the hematoxylin stain (2 g/l hematoxylin and 0.2 g/l potassium iodate ($KIO_3$) in distilled water) for 30 min and rinsed in three changes of distilled water for an hour. Photographs of root tips were taken using a stereo microscope fitted with an AxioCam ERc 5s imaging device and saved for later scoring of each root tip.

Means of the 4 seedlings per replicate were used for statistical analysis.

### Testing of acidity and aluminium response in pot experiments

According to the results of solution culture, accessions representing different combination of $Al^{3+}$-and acidity response (Table 2) were selected for further evaluation. The experiment was conducted twice, first on peat-based medium and again on perlite.

### Peat growing medium

The peat medium was prepared with nine parts peat (FPM 420 F6, Kekkilä Oy, Vantaa, Finland) to 1 part sand (v:v). The pots were 7.5 l in size and were sunk into sand to prevent overheating of the roots by direct sunlight. The average nutrient composition of the peat in g/m$^3$ basis included nitrogen 2000, phosphorus 500, potassium 2000, and limestone (Ca and Mg) 1.5. The natural pH of the medium was 4.5 and in one-third of the pots was adjusted to pH 7.0 by liming. Half of the pots at pH 4.5 were supplied with 3 l of 82 $\mu$mol/l

**Table 2** Summary of acid and aluminium responses of the 10 faba bean accessions chosen from the solution culture experiment for further investigation.

| | | Aluminium response | | |
| --- | --- | --- | --- | --- |
| | | Tolerant | Intermediate | Sensitive |
| Acidity response | Tolerant | NC 58, Dosha | | Kassa, GLA 1103 |
| | Intermediate | Gebelcho | | |
| | Sensitive | Aurora, Messay | Tesfa, EH 06006-6 | Babylon |

$Al_2(SO_4)_3.16H_2O$ solution at sowing, and all other pots were given 3 l of tap water bringing the medium to field capacity. Thus, there were three treatments, pH 7.0, pH 4.5, and pH 4.5 with $Al^{3+}$.

Plants were grown in the open air for 58 days from 28 May to 24 July 2015 at the University of Helsinki. Before sowing, seeds were inoculated with *Rhizobium leguminosarum* biovar. *viciae* (faba bean strain, Elomestari Oy, Tornio, Finland). Four seeds of each accession were sown directly in individual pots at a depth of 2 cm and a week after emergence they were thinned to 3 plants per pot. The experiment was terminated when plants started flowering.

### Perlite growing medium

Nutrient solution culture was prepared using a complete fertilizer (Superex Peat; Kekkilä Oy, Vantaa, Finland) containing 6.8 mmol/l $NO_3^-$, 0.9 mmol/l $NH_4^+$, 0.29 mmol/l urea, 1.1 mmol/l $H_2PO_4^-$, 6.7 mmol/l $K^+$, 1 mmol/l $Mg^{2+}$, 27.8 $\mu$mol/l $H_2BO_3^-$, 3.1 $\mu$mol/l $Cu^{2+}$, 48.3 $\mu$mol/l $Fe^{2+}$, 18.2 $\mu$mol/l $Mn^{2+}$, 0.52 $\mu$mol/l $MoO_4^{-2}$, 4.6 $\mu$mol/l $Zn^{2+}$, and 1 mmol/l $SO_4^{-2}$. To provide calcium, 2 mmol/l $CaCl_2$ was added. According to the pH requirement of the nutrient solutions for each treatment, the pH of the solution was adjusted using HCl and $Na_2CO_3$.

Plants were grown in an open-sided greenhouse cage for 42 days from 18 May to 30 June 2016 at the University of Helsinki. For each plant, two 3-l pots were stacked on top of each other in order to provide depth for rooting. Each pot was 20 cm deep and 15 cm in diameter, with four drainage holes that were 2 cm in diameter in the bottom pots and 3 cm in the top pots, so the impairment of root growth was minimal. The bottom pots were filled with 2 l of perlite and 1 l of fine sand on top of the perlite to hold the top pot firmly, and the top pots were filled with 3 l of perlite. Two seeds per top pot were planted and each pot was irrigated with tap water to field capacity. Plants were thinned to one per top pot after five days. For 10 days after sowing, pots were watered with 200 ml of tap water every other day. Thereafter, each pot received 200 ml of nutrient solution of the corresponding treatment every other day for 32 days. Treatments were implemented by fertigation with neutral pH (7.0) nutrient medium, acid medium at pH 4.5, and pH 4.5 + 82 $\mu$mol/l $Al_2(SO_4)_3.16H_2O$. The experiment was terminated when plants started flowering, even though this was fewer days than in the peat experiment.

## Experimental design

The peat and perlite experiments were a split-plot design, with four replicate blocks, three treatments (neutral, acid and aluminium) as the main plots and accessions as subplot.

## Data collected
### Peat experiment
The rate of leaf photosynthesis and stomatal conductance were measured using a LI-6400 Portable Photosynthesis System (LI-COR, Lincoln, NE, USA) at 1,400 $\mu$mol/m$^2$/s radiation level and adjusted cooling temperature of 25 °C, on 6 cm$^2$ leaf area in bright sunlight from 11:00 a.m. to 2:00 p.m. One leaf per pot was used, chosen from the middle of the shoot having been fully exposed to direct sunlight at 25, 35, and 45 days after sowing. Leaf chlorophyll concentration was measured using an optical chlorophyll meter, SPAD-502 (Minolta Camera Co, Ltd., Tokyo, Japan), on the same days. Two leaves per plant were measured and the average of the two was recorded. Data from 35 days after sowing are presented here.

At the end of the experimental period, 52 days after sowing, the leaves of each plant in the pot (three plants per replication) were severed from the stem and the total leaf area was measured using a LI-COR Model LI-3000A Portable Area Meter (LI-COR, USA) and the average values were calculated for each replicate. Plant roots were carefully removed from both the pot and the peat and scored for nodule quantity and quality. Nodule quantity was measured by two people scoring, zero for absent, 1 for a few nodules, 2 moderate quantity, and 3 for prolific nodule production. Similarly, nodule quality was also scored by two people as follows. "0" was given for absent, "1" was given when >50% of the nodules were white in color, "3" was given when >50% of the nodules were pink in color, "2" was given when each white and pink nodules were nearly equal by numerous. In both cases, the average of the two scores were taken as a value of each unit of plant per replication. Finally, the root and shoot dry weight were taken after drying the biomass in oven at 76 °C for 48 h. Samples of shoot and potting medium were taken for aluminium analysis.

### Perlite experiment
On day 41, stomatal conductance was measured using a Leaf Porometer (Decagon Devices, Inc., Pullman, WA, USA) once per plant. On day 42, canopy temperature was measured using a FLUKE Model 574 Precision Infrared Thermometer (Fluke Corporation, Everett, WA, USA) and leaf chlorophyll concentration was measured using the SPAD-502 as described above. Roots were carefully removed from the perlite and taproot length was measured with a ruler. The root and shoot dry weight were determined as described above. Root to shoot dry weight ratio was calculated by dividing the root weight by the corresponding shoot weight.

## Aluminium concentration analysis
Samples of shoot and growing media were milled and representative 250 mg sub-samples were digested with acid in a microwave oven (Mars Express, CEM, USA) conforming to EPA 3052 (9 ml 70% $HNO_3$ and 1 ml 30% $H_2O_2$ at 1,600 W and 175 °C; *EPA, 1996*). The concentration of Al was measured with inductively coupled plasma-optical emission spectrometry (ICP-OES), Thermo Scientific iCAP 6000 Series (Thermo Fisher Scientific, Waltham, MA USA). The limit for detection of Al by the ICP was 0.032 mg/l.

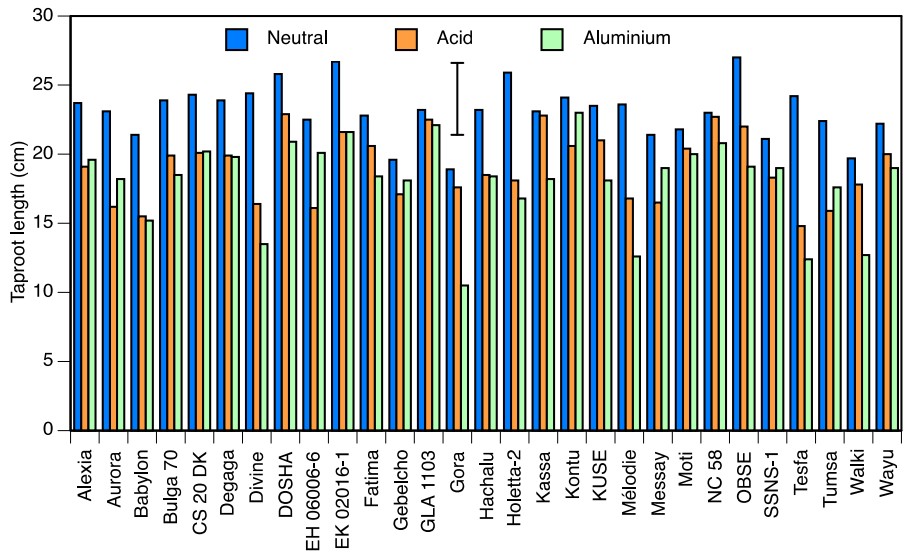

**Figure 1** **Taproot lengths in solution culture.** Taproot length of seedlings of 29 accessions of faba bean grown in solution culture for 10 days in neutral (pH 7.0) or acid (pH 4.5) nutrient solution, or 8 days in the acid solution followed by 2 days in aluminium-containing acid (pH 4.5, 82 μmol/l $Al^{3+}$) solutions. Error bar shows LSD (5%).

### *Data analysis*

Analysis of variance was conducted using SPSS version 22.0 (IBM Inc., Chicago, IL, USA) software package. Accession and culture media were treated as fixed effects and replicate as a random effect in the split-plot design. Treatment means were separated by LSD (5%).

## RESULTS

### Solution culture experiment

Taproot length at pH 7.0 ranged from 18 cm in Gora to 27 cm in Obse and EK02016-1, and at pH 4.5 from 14.8 cm in Tesfa to 22.9 cm in Dosha (both $P < 0.05$; Fig. 1). Accessions Kassa, NC58 and GLA1103, all showed less than 3% reduction in root length in acid condition, whereas 8 accessions showed significant reductions, with the greatest in Tesfa, Divine and EH06006-6 which showed 30–40% setback.

Two of the three aluminium treatments were relatively uninformative. The difference in mean root lengths between the 0 and 41 μmol/l aluminium treatments was not statistically significant (1.2 cm longer with aluminium than without). The 123 μmol/l treatment did not result in a significant difference in root length from the 82 μmol/l treatment, and very few of the roots regrew, so it appeared to be too strong a dose. Hence, only the 82 μmol/l treatment provided informative results and we present only those here.

Roots grown in aluminium-free solutions did not stain with haematoxylin (Fig. 2A). The minimum haematoxylin stain score in the 82 μmol/l treatment was found in Dosha, followed by Gebelcho, while the maximum was in Babylon, followed by Degaga ($P < 0.05$; Table 1). Babylon showed no apparent root regrowth following transfer to aluminium-free solution, whereas Hachalu showed the most vigorous regrowth ($P < 0.05$). Accessions

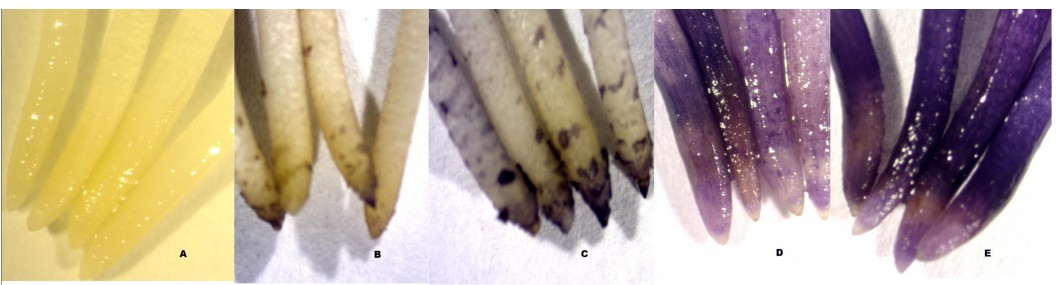

**Figure 2** **Haematoxylin staining of root tips of faba bean.** (A) Acid treatment without aluminium; (B–E) after 82 µmol/l $Al^{3+}$ treatment; (B) accession Dosha, score 0, $\leq 25\%$ stain; (C) accession Aurora, score 1, $25\% < x \leq 50\%$ stain; (D), accession Wayu, score 2, $50\% < x \leq 75\%$ stain; (E), accession Babylon, score 3, $>75\%$ stain.

EH06006-1, followed by Kontu, Messay, and Aurora, all produced longer roots in the 82 µmol/l aluminium treatment than in the acid treatments (Fig. 1), although these differences were not significant. Aluminium treatment caused significant reductions in root growth in Gora and Walki.

Summing the ranks of hematoxylin staining, root regrowth, and root length setback by aluminium (Table 1) indicated that Gebelcho, Aurora and Messay were the three most tolerant accessions, whereas Kassa, Babylon and Divine were the most susceptible.

Comparison of these results showed that acidity response and aluminium responses have some independence. On this basis, 10 accessions were chosen for further experiments (Table 2).

## Experiments in peat and perlite potting media

In the peat experiment, leaf area was significantly different in all three treatments, neutral > aluminium > acid, but leaf temperature did not show significant effects of treatment. Rate of photosynthesis increased in the order neutral < acid < aluminium, and shoot dry matter was greatest in the aluminium treatment. Nodule quantity, nodule quality and root dry matter were all highest in the neutral treatment with no significant difference between acid and aluminium treatments (Table 3). Similarly, shoot dry weight and root to shoot dry weight ratio were highest in neutral treatment. Chlorophyll concentration and stomatal conductance were significantly lower in the neutral treatment than in the other two, which did not significantly differ (Table 4).

In the perlite experiment, many results were similar. Chlorophyll concentration was again significantly lower in the neutral than in the other two treatments (Table 4). Babylon and GLA 1103 showed high leaf chlorophyll concentration in both acid and aluminium treatments. On the other hand, Kassa in the acid, and NC 58, Gebelcho and Messay in the aluminium treatment showed low chlorophyll concentrations (Fig. 3A). There was no significant difference in stomatal conductance or leaf temperature between treatments. Differences in root and shoot dry matter were also not significant. Root length, however, was significantly greater in the neutral treatment than the other two. In Aurora, Babylon, EH 06006-6, Messay and Tesfa, the setback due to acidity was greater than the LSD, but in the other 5 accessions the effect of acidity was very small (Fig. 3B). The effect of aluminium

**Table 3** Accessions and treatment means of leaf and nodule characters measured in either peat or perlite potting media, $n = 4$.

| Accession | Peat experiment | | | | Shoot Al (mg/kg) | Perlite experiment |
| --- | --- | --- | --- | --- | --- | --- |
| | Leaf area (cm²) | Photosynthesis (µmole/m²/s) | Nodule quantity | Nodule quality | | Canopy temperature (°C) |
| Aurora | 831 | 16.4 | 2.33 | 2.50 | 43.1 | 22.5 |
| Babylon | 722 | 18.6 | 1.75 | 2.42 | 44.1 | 22.8 |
| Dosha | 640 | 14.8 | 1.08 | 1.58 | 45.2 | 23.2 |
| EH 06006-6 | 551 | 15.1 | 0.83 | 1.08 | 68.6 | 23.7 |
| Gebelcho | 617 | 16.9 | 0.83 | 0.92 | 42.9 | 23.0 |
| GLA 1103 | 606 | 20.1 | 2.08 | 2.58 | 49.3 | 22.6 |
| Kassa | 533 | 16.5 | 1.08 | 1.33 | 50.8 | 22.9 |
| Messay | 547 | 17.2 | 1.67 | 2.33 | 59.5 | 22.7 |
| NC 58 | 569 | 14.7 | 1.00 | 1.17 | 79.1 | 22.8 |
| Tesfa | 475 | 18.3 | 0.92 | 1.33 | 72.8 | 22.2 |
| SE | 32 | 0.72 | 0.27 | 0.32 | 8.2 | 0.24 |
| LSD (5%) | 89 | 2.02 | 0.75 | 0.90 | 23.1 | 0.66 |
| Treatment | | | | | | |
| Neutral | 780 | 15.6 | 1.90 | 2.10 | 44.7 | 22.7 |
| Acid | 491 | 16.7 | 0.98 | 1.50 | 57.4 | 22.8 |
| Aluminium | 556 | 18.3 | 1.20 | 1.58 | 64.6 | 23.0 |
| SE | 17 | 0.39 | 0.15 | 0.18 | 4.49 | 0.13 |
| LSD (5%) | 49 | 1.11 | 0.41 | 0.49 | 12.7 | 0.36 |
| *P*-value | | | | | | |
| Treatment | *** | ns | * | ns | ns | ns |
| Accession | *** | *** | *** | *** | * | ** |
| Treatment × Accession | ns | ns | ns | ns | ns | ns |

**Notes.**

*, **, *** $p < 0.05$, 0.01, 0.001, respectively, ns (not significant) $p > 0.05$, SE is standard error.

on reducing root length was significant in Kassa, whereas Gebelcho, Dosha and Aurora showed slightly greater taproot length under aluminium stress.

Chlorophyll concentration in the two pot experiments showed similar effects of accessions except that Aurora was an outlier (Table 4). Without Aurora, the correlation coefficient of the other 9 accessions was 0.853 ($P < 0.01$). Stomatal conductance values in the two experiments were also highly correlated, again with the exception of Aurora, and with the same $r$ value. Root and shoot dry weights, however, were not significantly correlated between the two experiments. Aurora produced the largest shoot and second largest roots in peat, whereas EH 06006-6 did so in perlite.

## Shoot and growing medium Al concentration

There was no significant difference between treatment means in shoot aluminium accumulation (Table 3). However, there was a highly significant difference among accessions, with Babylon followed by Aurora showing low shoot Al concentration. Aluminium concentration in peat growing medium was far higher in aluminium-treated medium at 1,107 mg/kg than the other two (150 mg/kg neutral, 124 mg/kg acid).

**Table 4  Accession and treatment means of leaf, root and shoot traits measured in both peat and perlite potting media, $n = 4$.**

| Accession | Chlorophyll concentration (SPAD unit) | | Stomatal conductance (mol H$_2$O/m$^2$/s) | | Root dry weight (g) | | Shoot dry weight (g) | | Root to shoot dry weight ratio | |
|---|---|---|---|---|---|---|---|---|---|---|
| | Peat | Perlite | Peat | Perlite | Peat | Perlite | Peat | Perlite | Peat | Perlite |
| Aurora | 38.7 | 35.2 | 0.44 | 0.52 | 3.16 | 1.52 | 8.32 | 2.89 | 0.40 | 0.56 |
| Babylon | 40.8 | 39.2 | 0.50 | 0.29 | 3.48 | 1.45 | 6.83 | 3.55 | 0.52 | 0.42 |
| Dosha | 33.3 | 35.2 | 0.39 | 0.31 | 2.27 | 1.76 | 7.20 | 4.45 | 0.33 | 0.40 |
| EH 06006-6 | 36.2 | 38.4 | 0.37 | 0.20 | 1.89 | 2.12 | 6.55 | 5.08 | 0.30 | 0.42 |
| Gebelcho | 34.6 | 35.8 | 0.49 | 0.31 | 2.08 | 1.87 | 6.82 | 4.93 | 0.32 | 0.39 |
| GLA 1103 | 39.7 | 40.2 | 0.62 | 0.47 | 2.40 | 1.44 | 6.11 | 2.89 | 0.43 | 0.53 |
| Kassa | 34.8 | 34.2 | 0.50 | 0.37 | 1.75 | 1.57 | 6.31 | 3.85 | 0.28 | 0.41 |
| Messay | 34.3 | 36.1 | 0.47 | 0.29 | 1.88 | 1.52 | 6.40 | 4.42 | 0.30 | 0.34 |
| NC 58 | 34.7 | 36.1 | 0.41 | 0.28 | 2.03 | 1.52 | 7.08 | 4.40 | 0.30 | 0.35 |
| Tesfa | 35.4 | 34.8 | 0.58 | 0.36 | 1.61 | 1.66 | 5.11 | 4.82 | 0.34 | 0.35 |
| SE | 0.61 | 0.79 | 0.04 | 0.03 | 0.13 | 0.11 | 0.32 | 0.25 | 0.02 | 0.03 |
| LSD (5%) | 1.72 | 2.21 | 0.10 | 0.08 | 0.37 | 0.30 | 0.89 | 0.70 | 0.06 | 0.07 |
| Treatment | | | | | | | | | | |
| Neutral | 34.5 | 35.1 | 0.35 | 0.37 | 2.43 | 1.77 | 5.38 | 4.34 | 0.28 | 0.42 |
| Acid | 37.4 | 36.7 | 0.51 | 0.33 | 2.05 | 1.59 | 6.10 | 3.94 | 0.39 | 0.42 |
| Aluminium | 36.8 | 37.7 | 0.56 | 0.32 | 2.28 | 1.57 | 8.54 | 4.10 | 0.39 | 0.41 |
| SE | 0.33 | 0.43 | 0.02 | 0.02 | 0.07 | 0.06 | 0.17 | 0.14 | 0.01 | 0.014 |
| LSD (5%) | 0.94 | 1.21 | 0.06 | 0.05 | 0.20 | 0.16 | 0.49 | 0.39 | 0.03 | 0.04 |
| $P$-value | | | | | | | | | | |
| Treatment | ** | ** | * | ns | ns | ns | *** | ns | * | ns |
| Accession | ** | *** | *** | *** | *** | *** | *** | *** | *** | *** |
| Treatment × Accession | ns | ** | ns | ns | ns | ns | ns | ns | ns | ns |

**Notes.**

*, **, *** $p < 0.05$, 0.01, 0.001, respectively, ns (not significant) $p > 0.05$, SE is standard error.

## DISCUSSION

Great variation among accessions was observed in acidity tolerance ranging from below 3% to 40% setback in taproot growth. Similarly, accessions showed variable responses to aluminium treatments ranging from no apparent taproot regrowth to prolific root elongation during recovery after being exposed to aluminium treatment. Generally, accessions showed independent tolerance to acidity and aluminium treatments (Table 2). Accessions tending to tolerate both acidity and aluminium were found to be moderately tolerant. The independence of these responses to acidity and aluminium is in agreement with *Kidd & Proctor (2001)*, who showed that crop accessions separately adapt to H$^+$ and Al$^{3+}$ toxicity as a result of the difference in the nature of soil parent materials (organic vs mineral) where the accession originated.

In general, the results obtained in solution culture experiment were confirmed in the two pot experiments. Babylon, however, performed unexpectedly well in the peat experiment in comparison with the other experiments, and results of leaf traits of Aurora were outliers in the regression of the peat and perlite experiments. Part of this difference may be attributed

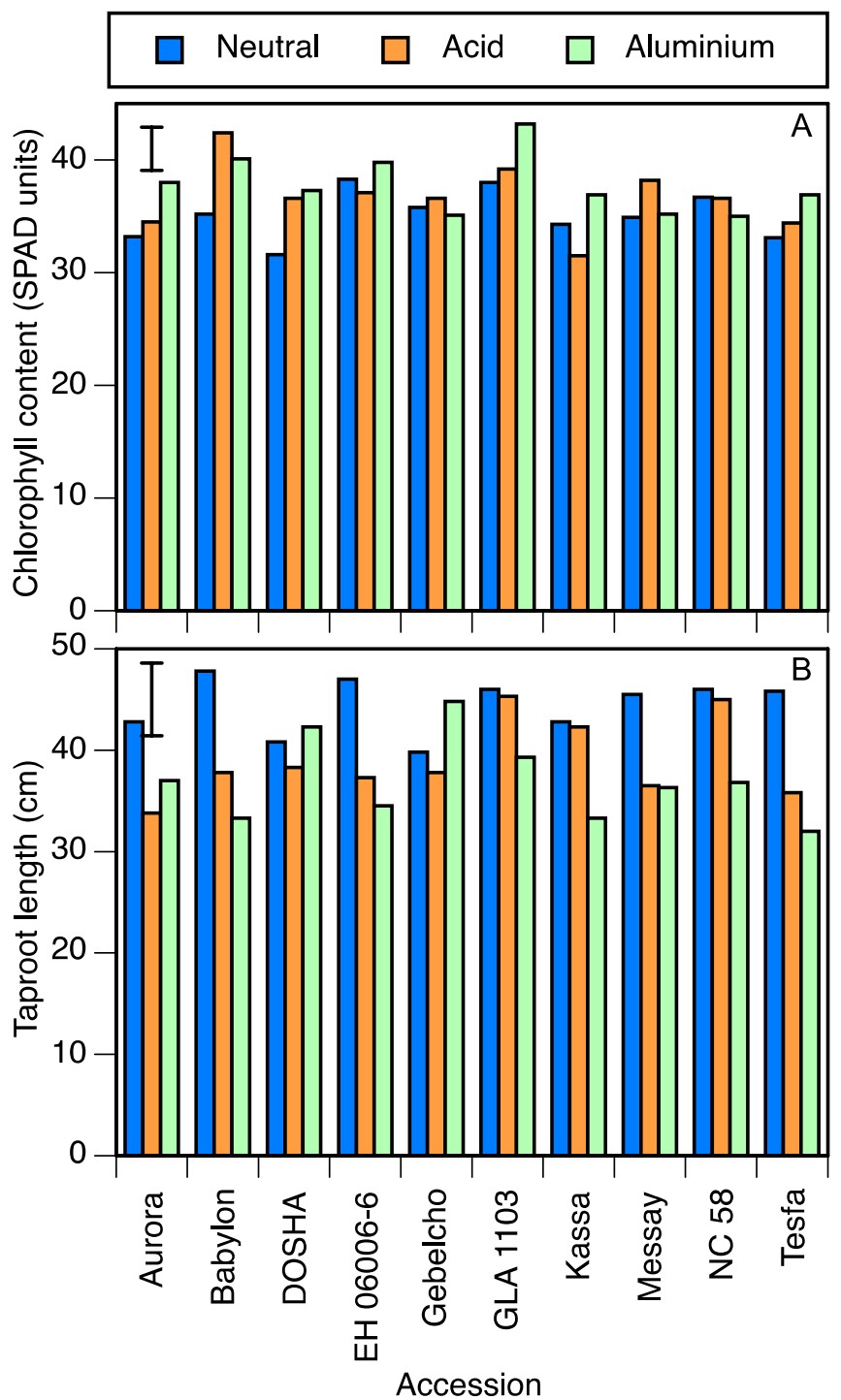

**Figure 3 Chlorophyll concentration and taproot length.** (A) Chlorophyll concentration (SPAD units) and (B) taproot length (cm) in 10 accessions of faba bean grown in perlite and fertigated with neutral (pH 7), acid (pH 4.5) or aluminium containing acid (pH 4.5, 82 $\mu$mol/l Al$^{3+}$) solutions. Error bar shows LSD (5%).

to heterogeneity of the samples, as the experiments were started with material as received, and part to specific responses of accessions to the rooting media. The heterogeneity was also shown in the highly variable responses to hematoxylin staining within some accessions. Nevertheless, results of the perlite experiment, conducted using third-generation inbred seed materials, showed great similarity with those of the solution culture experiment (compare Figs. 1 and 3B). Inbreeding of accessions to produce lines that are as homozygous as possible is well known to be necessary in cross-pollinated crops like faba bean.

The combination of reduced leaf size and increased chlorophyll concentration was indicative of stress due to either acidity or aluminium. Multiplying these two values together (e.g., from the treatment means in Tables 3 and 4) showed that even though the chlorophyll concentration was higher in the stress treatments, chlorophyll content per plant was lower. By these measures, certain accessions showed little stress from one or the other treatment, particularly in the perlite experiment, as confirmed by their root lengths and stomatal conductance values. Those accessions reported in solution culture experiment as tolerant in either of the stresses, such as Kassa in the acid treatment, showed lower chlorophyll concentrations in the tolerated stress treatment, whereas those accessions reported as sensitive to acid and/or aluminum stress, such as Babylon, showed higher concentrations (Fig. 3A). Similarly, GLA 1103 (aluminium sensitive) and NC 58 and Messay (aluminium tolerant) showed the highest and the lowest photosynthesis rates in aluminium treatment (Table 3), respectively. In certain accessions of wheat, a relatively low concentration (16 µmol/l) of Al increased chlorophyll a and b concentrations (*Alamgir & Akhter, 2010*), but 130 µmol/l decreased leaf chlorophyll concentration (*Ohki, 1986*).

Shoot Al concentration analysis indicated conflicting results among Al-tolerant accessions. The shoot Al concentrations of Aurora, Gebelcho and Dosha was low, whereas those of NC 58 and Messay were relatively high (Table 3), and the accession by treatment interaction was not statistically significant. Faba bean can exclude aluminium by chelation with exuded citrate (*Chen et al., 2012*), and this would be seen as stronger hematoxylin in susceptible accessions, as shown for wheat (*Polle, Konzak & Kittrick, 1978*). Its mechanism of Al tolerance may also include sequestration, as found in other species (*Osaki, Watanabe & Tadano, 1997*).

In terms of taproot growth along with root and shoot dry weights, accessions responded in several ways to the acidity and aluminum treatments. There was significant variation among treatments in taproot length (neutral > acid > aluminium) but not in root dry weight, showing that roots were shorter and thicker in stressed conditions that otherwise. Taproot length of tolerant accessions including GLA 1103, NC 58 and Kassa was not affected by acidity, whereas that of Aurora, Tesfa and Babylon was severely affected. Similarly, Kassa, Babylon, Tesfa and EH 06006-6 were greatly affected by the chosen concentration of aluminium, but the root lengths of Gebelcho, Dosha and Aurora were slightly longer under this stress (Fig. 3B). While this increase was not statistically significant, it was consistent with proposed root growth enhancement properties of aluminium under acidic conditions (*Kinraide, 1993*; *Osaki, Watanabe & Tadano, 1997*). In Al-stimulated plants, such as, *Acacia mangium* Willd. and some lines of *Oryza sativa* L. (*Osaki, Watanabe & Tadano, 1997*),

available $Al^{3+}$ at 44 $\mu$mol/l concentration applied as $Al_2(SO_4)_3$ was reported to increase the activity of roots for P uptake and initiation of many laterals.

The rhizotoxicities of $Al^{3+}$ and $H^+$, may partially alleviate each other, due to the competition of the positively charged cations for binding sites of the negatively charged epidermal layers of the roots (*Kinraide & Parker, 1987*; *Kinraide, Ryan & Kochian, 1992*). At low soil pH, $Al^{3+}$ has a stronger effect in amelioration of toxicity that is caused by cations such as $H^+$, $Na^+$, $K^+$, $Ca^{2+}$, and $Mg^{2+}$ (*Fawzy, Overstreet & Jacobson, 1954*; *Kinraide & Parker, 1987*; *Kinraide, Ryan & Kochian, 1992*; *Yan, Schubert & Mengel, 1992*; *Kinraide, 1993*; *Kinraide, 2003*). This reciprocal alleviation is probably the reason for the observed slight increase in taproot length and the development of smooth root surface (without callus marking) in some accessions treated with 41 $\mu$mol/l $Al^{3+}$ in the first experiment.

There was an important effect of pH but not of aluminium on nodule quantity and quality. Low pH reduced nodule mass by 48% and nodule quality by 29%. Similarly, acidity significantly decreased nodule numbers (as much as 20–33%) and activity in red clover inoculated by two different bacterial strains (*Shirokikh, Shirokikh & Ustyuzhanin, 2005*). Decreased plant dependence on fixed nitrogen was reported in subterranean clover (*Trifolium subterraneum* L.) growing in low soil pH and high soil aluminium (*Unkovich, Sanford & Pate, 1996*). *Correa, Aranda & Barneix (2001)* reported that alfalfa (*Medicago sativa* L.) and *Lotus glaber* Mill. varied in their nodulation and nitrogen fixation ability at pH 4.0 due to variation in host and/or strain tolerance to low pH. In this study, only one inoculum was used, so we were unable to test the effect of its genotype, but it is known that rhizobium symbionts of faba bean differ in their sensitivity to low pH, and their sensitivity to Al concentration depends on the other ions in the solution (*Kinraide & Sweeney, 2003*; *Bayoumi Hamouda et al., 2009*).

## CONCLUSION

Responses to acidity and aluminium toxicity were independent. A concentration of 82 $\mu$mol/l aluminum sulfate [$Al_2(SO_4)_3.16H_2O$] was suitable for discriminating the sensitivity of faba bean accessions, whereas 41 $\mu$mol/l had too little effect on root growth and 123 $\mu$mol/l was too much. Taproot length, root length setback by treatment and root regrowth procedures were found to be sound methods for preliminary screening of accessions for acidity and aluminium tolerance. At a later stage, taproot length, root and shoot biomass coupled with chlorophyll concentration and stomatal conductance provided reliable discrimination between accessions. Results, generally agreed between experiments, with some differences attributable to uniformity of seed and to specific responses of rooting medium.

High throughput screening for acidity and Al tolerance at an early stage of seedling growth was possible in the solution culture growth medium. Nevertheless, for further screening of faba bean accessions for an extended growth period, perlite was found to be an appropriate medium owing to its inert nature, free drainage allowing changes of treatment medium, provision of support for plants allowing monitoring of leaf and shoot responses to the acidity and aluminium stresses, and easy extraction of entire root systems for evaluation without encumbering medium.

Kassa, GLA 1103, Aurora, Messay, NC 58 and Babylon were selected for future breeding, genetics and physiological studies on Al tolerance in faba bean.

## ACKNOWLEDGEMENTS

We acknowledge Gemechu Keneni for his help during seed collection in Ethiopia and Marjo Kilpinen, Markku Tykkyläinen, Jouko Närhi and Sanna Peltola for their support in the laboratory and greenhouse activities during the conduct of the research, and Mikko Lehtonen for his assistance with microscope imaging at the University of Helsinki, Finland. Finally, we also thank Pertti Pärssinen for his valuable comments and remarks during the development of the manuscript. We extend our thanks to the Holeta Agricultural Research Centre for providing us 20 Ethiopian faba bean accessions and the Ethiopian Biodiversity Institute for its support to export the samples from Ethiopia to Finland.

### Funding

This work was supported by the Finnish Cultural Foundation and the Center for International Mobility. The funders had no role in study design, data collection and analysis, decision to publish, or preparation of the manuscript.

### Grant Disclosures

The following grant information was disclosed by the authors:
Finnish Cultural Foundation.
Center for International Mobility.

### Competing Interests

The authors declare there are no competing interests.

### Author Contributions

- Kiflemariam Y. Belachew conceived and designed the experiments, performed the experiments, analyzed the data, wrote the paper, prepared figures and/or tables.
- Frederick L. Stoddard conceived and designed the experiments, analyzed the data, contributed reagents/materials/analysis tools, prepared figures and/or tables, reviewed drafts of the paper.

### Data Availability

 The raw data has been supplied as a Supplementary File.

### Supplemental Information

Supplemental information for this article can be found online at http://dx.doi.org/10.7717/peerj.2963#supplemental-information.

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
