# Peer review of "Screening of faba bean (Vicia faba L.) accessions to acidity and aluminium stresses"

_PeerJ, doi:10.7717/peerj.2963_

## Round 0.1 · original submission · Minor Revisions

· Academic Editor

Minor Revisions

While this is generally a solid piece of work, there are some deficiencies noted by the reviewers that must be addressed before it is suitable for publication. Several of these are typos, or involve minor rewrites to include or clarify pertinent information. Please address these in your revised manuscript and response letter.

Of greatest importance is that you address the shortcomings in the statistical treatment of the results. Please indicate clearly the tests applied and provide results and level of replication. The details of the test can be placed in the supplementary information but you do need to include enough information that the reader can be confident that the tests have been applied correctly.

There seems to be a contradiction in your discussion in that you say there was no correlation between Al and acid tolerance, but also that "[o]utstanding tolerance to either of the treatments...was found to relate with sensitivity for the other". Please ensure that you resolve this apparent contradiction in your revised manuscript.

Finally, Reviewer #2 makes some insightful comments regarding the relative ratings of the tests for Al tolerance that I hope you will consider when making your revisions. This is not a condition of publication, but I think it would be useful to at least mention the underlying assumptions of the three tests.

Reviewer 1 ·

Basic reporting

The article meets the standard of Peer J.

Experimental design

The article clearly defines the research question and has been conducted rigorously, but some methods were not described with sufficient information. For example, the size of perlite and fine sand and pore diameter of grainage holes were not given. And the used statistic method were not clearly discribed.

Validity of the findings

The data were robust and controlled,and were well discussed. The findings that a concentration of 82 μM aluminum sulfate was suitable for discriminating the sensitivity of faba bean accessions, that perlite was suggested to be an appropriate medium for further screening of faba bean, and that six accessions selected for future studies on Al tolerance in faba bean, were insteresting.

Additional comments

Dear Authors,
In reference to manuscript "Screening of faba bean (Vicia faba L.) accessions to acidity and aluminium stresses" reference PeerJ 14133 sent for review, I consider that the theme is appropriate and within the scope of your journal, but there are many errors and shortcomings in the work. In order to help improve the manuscript, I suggest the authors the following:
Aluminium content analysis: Limit of detection of the metal must be added.
Data analysis: The statistics method for differences analysis should be given with a brief introduction
Uniform the concentration units, μmol/L and mmol/L are better than μM and mM.
Line 120, correct the multiplication sign in 78 cm x 56 cm x 18 cm.
Line 160, did the growing condition of pH include 4.5, 7.0 and 4.5 with aluminium treatment, as those in perlite medium? Reworded this sentence. Rewrite the sentence.
Line 163, the number of days was incorrect. And why the growth time for peat-based medium and perlite medium were different?
Line 176, 178, provide detail information for perlite, fine sand, and drainage hole, such as their sizes or diameter, and the depths for root growth.
Line209, why the drying temperature was not 70 °C as usual?
Line 236-238, what is the difference between 0, 41 μM and 82, 123 μM aluminium treatments?
Line 326-327, what is the exact meaning of “there was no significant accession by treatment effect”, rewrite the sentence.
Give the number for the replication in all tables.
There was no so-called data for perlite media in Table 3. What data did the standard error calculated from, and what was the number of the samples for statistics?
There was no “****” in Table 3 and 4.

Reviewer 2 ·

Basic reporting

Identifying acid soil tolerant pulses like faba bean is an important objective for agriculture. This study goes some way to develop methods for screening genotypes for aluminium (Al) tolerance to identify useful material for breeding programs. A number of different tests are compared and results compiled to provide overall ranking for tolerance. This involved some root traits as well as canopy traits. While this study appears to be solid and performed competently it lacks novelty and provides moderate interest only. The authors are encouraged to go the next useful step in future studies. This involves making crosses between resistant and sensitive genotypes to generate segregating populations which can be used for examining the genetics of this trait. This strategy would help answer questions like how many genetic loci are likely to be involved in resistance etc. It could also provide markers for marker assisted breeding.

Comments and suggestions below:
(1) The description of the treatment in the hydroponics experiments were confusing. A lot of different treatments were included in the one experiment and as a result it became unclear what replicates were going where and for how long each of the treatments were. I suggest the authors revise this part to make this very clear – at least make the duration of each treatment clear.
(2) Footnotes to Table 3 and 4 are showing four **** when in fact only three *** are required.
(3) Figure 3 is not really data and could be included in a Supplemental section.
(4) More comments to explain the inverse relationship between the chlorophyll and Al resistance would be welcomed.

Experimental design

no comments

Validity of the findings

(5) Line 243: States that the listed cultivars had longer roots in the Al treatment than the low pH without Al treatment. This is not strictly true. If the values are not statistically different then they cannot be distinguished from one another.

(6) Are the data in Figure 1 and Figure 4 all analysed with a single 2-factor ANOVAR? If so these results should be shown in a Supplementary table.

(7) Although a LSD bar is included in the Figure 4 we are not told what is actually different to what?

Additional comments

(8) Table 1 has a problem which I would like to explain in detail. It is true that for many other species Al resistance is correlated with Al exclusion from root apices and this is easily scored with haematoxylin staining of root apices. In those cases haematoxylin staining is a good surrogate for Al resistance. When the mechanism of resistance is unknown we cannot assume that exclusion is the mechanism of resistance and therefore it is unclear whether haematoxylin staining is a useful test. Table 1 compiles three different tests for Al tolerance and gives an overall score and ranking. The final scores are equally contributed by haematoxylin staining, regrowth and relative root growth. I would argue that the latter two are far more important than the haematoxylin staining because some species plants can accumulate Al and not show ill effects. Unless you know with confidence that Al resistance is due to Al exclusion from the root apices, then perhaps the contributions of the root growth measurements should be given relatively more importance in the final ranking than the haematoxylin staining. We do not really mind if roots stain darkly in haematoxylin as long as they grow well in Al solution. Root growth is the most direct measure of resistance.

---

## Round 0.2 · accepted · Accept

· Academic Editor

Accept

Thank you for addressing the questions raised by the reviewers of the original submission. I am pleased that you found the additional comments from Reviewer 2, regarding the interpretation of the root growth and haemotoxylin staining tests, to be useful.

Reviewer 1 has made some further minor observations that you are free to address or not as you prefer in the final manuscript.

Reviewer 1 ·

Basic reporting

1. Line 15-16,revised as where acidity and aluminium are problems or not.
2. The unit of liter would be better express as capital letter, L, instead of little letter, l.
3. Line 241, use the full abbreviation of ICP-OES.
4.Line 403, Whose responses?

Experimental design

No comment.

Validity of the findings

No comment.